# Regulating Immune Responses Induced by PEGylated Messenger RNA–Lipid Nanoparticle Vaccine

**DOI:** 10.3390/vaccines13010014

**Published:** 2024-12-27

**Authors:** Hyein Jo, Jaewhoon Jeoung, Wonho Kim, Dooil Jeoung

**Affiliations:** Department of Biochemistry, College of Natural Sciences, Kangwon National University, Chuncheon 24341, Republic of Korea; qnfdudn1212@gmail.com (H.J.); heyjhw@kangwon.ac.kr (J.J.); kimwonho99@kangwon.ac.kr (W.K.)

**Keywords:** ABC phenomenon, allergies, complement, immune tolerance, innate immunity, mRNA-LNPs, PEG

## Abstract

Messenger RNA (mRNA)-based therapeutics have shown remarkable progress in the treatment and prevention of diseases. Lipid nanoparticles (LNPs) have shown great successes in delivering mRNAs. After an mRNA-LNP vaccine enters a cell via an endosome, mRNA is translated into an antigen, which can activate adaptive immunity. mRNAs can bind to various pattern recognition receptors (PRRs), including toll-like receptors (TLRs), and increase the production of inflammatory cytokines. This review summarizes mechanisms of innate immunity induced by mRNAs. Polyethylene glycol (PEG) has been employed as a component of the mRNA-LNP vaccine. PEGylated nanoparticles display enhanced stability by preventing aggregation of particles. However, PEGylation can cause adverse reactions, including blood clearance (ABC) of nanoparticles via complement activation and anaphylaxis. Mechanisms of PEG-induced ABC phenomenon and anaphylaxis are presented and discussed. There have been studies aimed at reducing immune responses associated with PEG to make safe and effective vaccines. Effects of modifying or replacing PEG in reducing immune responses associated with PEGylated nanoparticles are also discussed. Modifying mRNA can induce immune tolerance, which can prevent hypersensitivity reactions induced by PEGylated mRNA-LNP vaccines. Current progress of immune tolerance induction in association with mRNA-LNP is also summarized. This review might be helpful for developing safe and effective PEGylated mRNA-LNP vaccines.

## 1. Immune Response Induced by mRNA Vaccines: Antigen Translation

Messenger RNA (mRNA) has been widely applied in therapies, including protein replacement therapies and cancer immunotherapies [1,2,3]. Successful development of COVID-19 vaccines has greatly contributed to the advancement of mRNA technology.

mRNA can be produced by in vitro transcription (IVT). It can then be translated into functional proteins. mRNA vaccines make it possible to mount an immune response without ever exposing to pathogens. mRNA vaccines show various advantages, including rapid preparation, reduced contamination, no induction of insertional mutagenesis, and high biodegradability [4]. These advantages make it possible to develop mRNA-based personalized vaccines. mRNAs have shown translational efficiency, reliable stability, and controlled immunogenicity [5]. It has been shown that mRNAs can bind to receptors and signaling pathways such as toll-like receptors (TLRs) and the JAK-STAT pathway and promote adaptive immune response [6]. Induction of adaptive immunity involves activation of T cells and the generation of specific antibodies.

Non-replicating mRNA (NRM) and virally generated, self-amplifying RNA (SAM) are now being investigated as vaccines. SAMs encoding a viral replication machinery allow for abundant production of antigen of interest [7]. mRNA structures include 5′ and 3′ untranslated structure regions (UTRs) [8]. 5′ UTR can be optimized to enhance translational efficiency [9]. Replacing ORF coding for viral structure proteins with viral RNA-dependent RNA polymerase can result in cytoplasmic expansion of the replicon structure. Refining 5′ cap structure, 3′ poly (A) tail, and codons can enhance the stability of mRNA [10,11].

mRNA vaccines encoding proteins of interest can be introduced into the cytoplasm of host cells, where they can be expressed into antigens [12]. Internalization of mRNA by antigen-presenting cells (APCs) can be made easier by performing nucleotide modifications and codon optimizations [5]. A single vaccination of mRNA can produce many antigens and induce major histocompatibility complex (MHC)-mediated T cell responses and production of neutralizing antibodies [13]. APCs can internalize the mRNA vaccine via endocytosis. mRNAs can escape endosomes and access the cytoplasm for translation into proteins of interest. The proteasome complex can break down intracellular proteins into antigenic peptides. Antigenic peptides are presented to CD4^+^ T cells or CD8^+^ T cells via MHC molecules on surfaces of APCs. Antigenic peptides can induce differentiation of CD4^+^ T cells into effector T cells, including T helper (TH) cells and T follicular helper (TFH) cells. TFH cells can promote germinal center reactions to induce the production of neutralizing antibodies. Activated CD8^+^ T cells by MHC-I/Peptide can exert cytotoxic effects. mRNAs can produce extracellular or cell surface proteins that can be recognized by cognate B cells and induce the production of neutralizing antibodies. Targeted delivery and endosomal escape remain challenging issues for developing mRNA-based therapy. Figure 1 shows that proteins encoded by mRNA can induce both innate immunity and adaptive immunity.

## 2. Structure of mRNA-LNPs

Lipid nanoparticles (LNPs), a leading non-viral delivery system, can efficiently co-deliver vaccines and immune adjuvants to lymphoid organs [14]. LNPs have shown successes as delivery vehicles for small molecules such as mRNAs and siRNAs [15,16]. Negative charges and toxicity of RNAs hinder their efficient uptake by host cells. LNP-encapsulation has been employed to generate safe and effective vaccines. LNPs can enhance the stability of messenger RNA and facilitate endosomal escape [17]. Enhanced half-life stability of LNPs can promote an enhanced permeation and retention (EPR) effect that can result in accumulation of LNPs in cancer tissues [18].

LNPs mostly contain four kinds of lipids: (1) cationic or ionizable cationic lipids such as N-[1-(2,3-dioleyloxy)propyl]-N,N,N-trimethylammonium chloride (DOTMA); (2) phospholipids such as dipalmitoylphosphatidylcholine (DPPC); (3) lipid-anchored polyethylene glycol (PEG) such as DMG-PEG2000; and (4) cholesterol [19]. Ionizable liposomes are neutral in the bloodstream. They display minimal off-target interactions with anionic cell membranes of blood cells [20]. In an endosomal environment, ionizable lipids undergo protonation on the liposome surface, which can promote membrane disruption and endosomal escape of mRNA. PEG-lipids can enhance the stability of mRNA-LNP by decreasing particle aggregation without causing changes in mitochondrial metabolism in neuroblastoma cells [21]. Decreasing PEG molarity or replacing PEG structures can increase protein expression by increasing the size of mRNA-LNP [22]. Large nanoparticles that contain more lipids and mRNAs can promote high endosomal escape and provide more mRNA for protein production. PEG molarity might affect innate and/or adaptive immunity by regulating the production of antigenic epitopes after vaccination with an mRNA-LNP vaccine. It is also probable that decreasing PEG molarity can change the physicochemical properties of nanoparticles to affect protein expression.

## 3. Induction of Immunity by Lipid Nanoparticles

mRNA-LNPs can act as an immune adjuvant based on the following findings: (1) mRNAs can be sensed by Toll-like receptor (TLR), melanoma differentiation-associated *protein* 5 (MDA5), and nucleotide-binding domain, leucine-rich–containing family, pyrin domain–containing-3 (NLRP3) [23,24,25]; (2) mRNA-LNPs can increase the production of inflammatory cytokines including IL-1α, IL-1β, and IFN-β, and IFN-γ through RNA sensing receptors [26,27,28,29]; and (3) mRNA-LNPs can promote adaptive immunity involving activation of T cells and germinal center (GC) B-cell responses [30,31,32]. BNT162b2 mRNA-LNP vaccine can increase levels of proinflammatory cytokines including IL-2, CCL2, CCL4, and CCL5 [33]. The BNT162b2 mRNA-LNP vaccine can also induce CD8^+^ T cell responses via type I interferon (IFN)-dependent MDA5 signaling [33]. In addition, ionizable lipids can activate the TLR2, TLR4, and NLRP3 inflammasome [34,35]. Cationic or ionizable cationic lipids and PEGylated lipids of the mRNA-LNP-COVID19 vaccine can also induce acute inflammatory syndrome by increasing the secretion of cytokines, including IL-1α, IFN-γ, IL-1β, and IL-8 by activating the complement system [34,36,37].

## 4. Innate RNA Sensing

PRRs are known to mediate innate immune responses [38]. PRRs are expressed on APCs such as monocytes and dendritic cells (DCs) [39]. PRRs include TLRs, leucine-rich repeat-containing receptors (NLRs), and retinoic acid-inducible gene 1 (RIG-I) like receptors [38,40]. TLRs are crucial mediators and regulators of host immunity [34].

Nucleic acids can act as molecular patterns [38]. Sensing IVT mRNA by endsomal TLR7/8 [41] can activate the myeloid differentiation factor (MyD) 88 pathway to initiate type1 IFN pathways [41,42]. TLR7/8 agonist can induce production of type1 IFN and IL-6 and adaptive immune responses [41]. Anticancer mRNA-LNP vaccines can trigger TLR4 [43], TLR7 [44], and can be stimulators of IFN gene 16 (STING16) signaling pathways [45]. RNA sensing by PRRs can also activate the IFN pathway, which in turn can increase the production of proinflammatory cytokines, resulting in the activation of APCs [46]. mRNA sensing by TLRs (TLR3, TLR7, and TLR8) can activate the innate immune system to increase production of proinflammatory cytokines [47]. Endosomal TLRs, specifically TLR3, TLR7, and TLR8, can act as viral RNA PRRs. Single-stranded RNA (ssRNA) can bind to TLR-7 and TLR-8 to activate the MyD88 pathway and induce production of the germinal center B cell-dependent IgG antibody [48]. TLR7 can activate NF-κB, MyD88, IRF7 pathways and increase secretion of CCL2, CXCL10, IL-1β, IL-6, IL-8, and type I IFNs [49,50,51]. TLR-3 can sense double-stranded RNA (dsRNA) and activate the TRIF (TIR-domain containing protein inducing type 1 IFN) pathway while decreasing the expression of adenosine deaminase RNA specific 1 (ADAR1) [52]. TLR-3 can also activate IRF3 and NF-kB to increase production of proinflammatory cytokines and type I IFN [53]. LNP-poly(I:C) can activate lysosomal TLR3 in human and mouse fibroblasts [54].

Viral infection can activate RNA sensors, including RIG-I-like receptors (RLRs), TLRs, protein kinase R (PKR), and ADAR1 [55]. RLRs can sense cytosolic RNA and act as sensors of the BNT162b2 vaccine [33]. Cytosolic RLRs and MDA5 along with mitochondrial antiviral signaling protein (MAVS) can increase production of proinflammatory cytokine [56,57]. DsRNA can activate MAVS and STING-mediated signaling pathways in response to ionizing radiation [56]. MDA5 and RLR can increase the secretion of proinflammatory cytokines by activating IFN response factor 3 (IRF3) and NF-kB [57]. Understanding mechanisms of innate immunity mediated by the IFN pathway is needed to develop safe and effective mRNA-LNP vaccines. Figure 2 shows that RNA sensing by innate PRRs can induce production of proinflammatory cytokines.

## 5. Translational Inhibition of mRNA by dsRNA

Innate RNA sensors can decrease the expression of antigen from mRNA by activating ribonuclease L (RNase L) that cleaves single stranded RNA (ssRNA) via JNK and p38 MAPK pathways [58]. RNase L can decrease the translation of stress-responsive genes by inducing ribosome stalling through activation of human leucine zipper and sterile alpha motif-containing kinase (ZAK) [58]. Double stranded RNA (dsRNA) produced during IVT can restrict mRNA translation by activating PKR and NF-kB [59,60]. PKR and RIG-1 are known to recognize different types of RNA to interfere with translation [61,62,63]. PKR can induce phosphorylation of eukaryotic initiation factor 2 (eIF2), which in turn can suppress mRNA translation [63,64].

DsRNA can also trigger innate immunity via oligoadenylate synthetase 3 (OAS) [65]. Upon detecting viral dsRNA, OAS3 can activate RNase L [66], which cleaves both cellular and viral RNAs, further triggering RLR signaling to increase the production of proinflammatory cytokines [67]. RNA-binding protein E3 (E3L) can bind to PKR and inhibit activation of PKR [68]. Thus, E3L can be employed to prevent translational inhibition by dsRNA. This can be used to develop effective mRNA-LNP vaccine. Figure 2 describes the mechanism of RNA degradation by PKR and OAS. To make effective mRNA-LNP vaccines, it is necessary to purify IVT mRNA without dsRNA.

## 6. Effects of Modifications of mRNAs on Immunogenicity and Translation

mRNA transfection can stimulate TLR3 and TLR7 and induce IFN-β expression in human and mouse fibroblasts [69]. Unmodified mRNA-LNP vaccines can activate innate immune responses that often lead to cytokine storms. Ss RNA 40, a synthetic mimic of SARS-CoV-2 RNA, can increase the expression of proinflammatory cytokines in monocyte-derived dendritic cells (MDDCs) via TLR8 [70]. Inositol requiring enzyme 1 α (IRE1α) inhibitor MKC8866 can prevent ssRNA40 from increasing the production of proinflammatory cytokines [70]. Thus, the TLR8-IRE1α system might serve as a target to control the cytokine storm associated with COVID-19 mRNA-LNP vaccines. TLR signaling can activate IRF3 to induce production of proinflammatory cytokines including IFN [71]. DsRNA can increase the expression of IFN stimulated genes (ISGs) by activating IRF3 and NF-κB [57].

Nucleoside-modified mRNA-LNP vaccines might abrogate innate immunity critical for inducing adaptive immunity. Therefore, it is advisable to modify components of mRNA-LNP vaccines. Post-transcriptional modifications such as RNA pseudouridylation can affect gene expression and regulate various biological processes [72]. Replacing uridine with pseudouridine (Ψ) or other derivatives such as N1-methylpseudouridine (m1ψ), 2-thiouridine, 5-methyluridine, and 5-methylcytidine can prevent endosomal TLR3 and TLR7/8 and cytosolic RIG-1 from recognizing mRNAs and increasing the production of IFN-β [69,73]. Thus, modifications of mRNAs can reduce innate immunity induced by mRNA-LNP vaccines. RNA-editing enzyme ADAR1 (adenosine deaminase acting on RNA) can inhibit immune checkpoint blocker (ICB) responsiveness by suppressing immunogenic dsRNAs arising from dysregulated expression of endogenous retroviral elements (EREs) [74].

Pseudouridylation of alpha ketoglutarate-dependent dioxygenase (ALKBH3), a tumor suppressor gene, can enhance the translation of ALKBH3 and reduce tumor growth [72]. Chemical modification of nucleotides can enhance translation levels by preventing innate sensors from recognizing IVT mRNA [75,76]. Enhanced protein translation efficiency can be attributed to stability of pseudouridine-modified IVT mRNAs resulting from a reduced activity of RNA-dependent PKR [75,76].

## 7. PEGylated mRNA-LNPs Cause Hypersensitivity Reactions, Allergies, and Complement Activation-Related Pseudoallergy (CARPA)

Safe and effective vaccines can stimulate the innate immune system and prime adaptive immune responses. However, immune-related adverse effects can arise because of immunological actions induced by these vaccines. Stimulation of RNA sensors such as TLRs can induce cytokine storm, airway infiltration of immune cells, and activation of mast cells [77]. It has been shown that mRNA COVID-19 vaccines can cause allergies [78,79]. COVID-19 can activate mast cell-derived proteases and increase levels of eosinophil-associated mediators based on assays employing sera and lung tissues of COVID-19 patients [77]. The BNT162b2 COVID-19 vaccine can exacerbate asthma by enhancing sensitivity to histamines in a human ex vivo model [80]. Phospholipase A2IIA activity is correlated with severity of COVID-19 vaccine [81]. The COVID-19 vaccine can increase levels of lysophosphatidic acid (LPA) and platelet activating factor (PAF) by phospholipase A2IIA [81]. It is well known that PAF mediates allergic reactions via the PAF receptor [82].

PEGylated proteins along with complete Freund’s adjuvant can induce production of PEG-specific antibodies [83]. PEG displays immunogenicity when it is used as an excipient in LNPs [84]. PEG can cause anaphylaxis in some drug reactions [85]. High molecular weight PEGs are known to cause allergic reactions. In animals, anti-PEG immunity consists of mostly the anti-PEG IgM response. PEGylated mRNA-LNP vaccines can induce the production of anti-PEG IgG, anti-PEG IgM, and anti-PEG IgE [86,87,88]. COVID-19 mRNA-LNP vaccines (Comirnaty and Spikevax) can induce hypersensitivity reactions (HSRs) or anaphylaxis by increasing levels of anti-PEG IgG/IgM [86]. Anti-PEG antibodies can be produced without prior exposure to PEGylated nanoparticles [89]. Anti-PEG Abs have been shown to be correlated with a reduced clinical efficacy of PEGylated therapeutics due to the presence of anti-PEG antibodies.

Anti-PEG antibodies on the surface of PEGylated nanoparticles can induce HSRs such as hypothermia and hypotension by activating Fcγ receptors on innate immune cells [90]. TLRs play a key role in allergic airway inflammatory responses, including airway infiltration of immune cells, increased levels of Th2 cytokines, and metaplasia of lung epithelial cells [91]. The expression levels of inflammation-related genes are decreased in TLR2 KO cells compared to those in wild type cells [91]. The TLR2-ERK signaling pathway can mediate allergic airway inflammation by regulating expression of Gal-3 [92]. The PEGylated siRNA-lipoplex can increase production of anti-PEG IgM by activating the TLR7 signaling pathway [93]. PEGylated liposomes containing TLR agonists can cause acute HSRs by inducing high levels of the anti-PEG antibody [94]. Thus, TLR signaling pathways might be responsible for the induction of allergic symptoms by PEG-mRNA-LNP vaccines.

People who have high levels of anti-PEG Ab in their blood can develop HSRs/anaphylaxis in response to PEGylated vaccines [95]. PEGylated proteins can induce production of anti-PEG antibody by activating T cells responses, while PEGylated liposomes function in a T cell-independent manner [96]. PEG has been implicated in pseudoallergic or anaphylactoid reactions via complement activation-related pseudoallergy (CARPA) [97]. Several types of anti-PEG antibodies are known to contribute to HSRs and premature drug release from PEGylated carriers [95,98,99]. Comirnaty, a PEGylated mRNA vaccine, can induce production of anti-PEG IgG and anti-PEG IgM antibodies, which in turn can activate the complement system to cause pseudoanaphylaxis [95]. Anti-PEG IgG, but not anti-PEG IgM, can induce symptoms of HSRs including hypothermia, altered lung function, and hypotension after administration of PEGylated liposomal doxorubicin (PLD) in C57BL/6 and non-obese diabetic/severe combined immunodeficiency (NOD/SCID) mice [90]. Anti-PEG IgG can cause HSRs through Fcγ receptors independent of complements [90]. Immune cells including basophils, monocytes, neutrophils, and mast cells can mediate anti-PEG IgG-mediated HSRs by recognizing mRNA-LNPs through PRRs such as nucleotide-binding oligomerization domain (NOD)-like receptors (NLR) [100]. This suggests that pathogenesis of allergic diseases involves both innate and adaptive immune responses. Anti-PEG IgE can also mediate HSRs induced by PEGylated nanoparticles [101].

Immunization with the mRNA-LNP COVID-19 vaccine can induce anaphylaxis, like CARPA, following their first or subsequent vaccinations [102,103]. CARPA results from elevated levels of complement C3a and sC5b-C9 [102]. Unlike other IgE-mediated reactions, CARPA can cause anaphylaxis upon first exposure. Allergic inflammations such as asthma involve lectin-dependent activation of the complement system via Ficoli-1 in response to allergens [104,105]. The mRNA-LNP COVID-19 vaccine can activate complements and increase the production of proinflammatory cytokines including IL-1α, IL-1β, IL-6, IL-8, IFN-γ, and TNF-α in PBMC cultures [36]. The COVID-19 vaccine can activate complements in an anti-PEG IgG-dependent manner and induce activation of basophils and M1 polarization of monocytes [78]. Thus, activation of complements caused by the COVID-19 vaccine can increase production of proinflammatory cytokines. PEGylation can reduce clearance of LNPs by mononuclear phagocytic cells by affecting binding of opsonin protein to liposomes [106,107,108,109]. In the case of the COVID-19 vaccine, interaction with immune cells can activate the complement system [110]. This implies that the stability of COVID-19 vaccine can be reduced by the complement system. Anti-PEG antibodies can reduce stability of PEGylated nanoparticles [111]. Anti-PEG antibodies can activate complement system by binding to PEG on the surface of PEGylated liposomal doxorubicin (PLD) [112]. Repeated injections of PEG-LNPs can decrease stability of PEG-LNPs [113]. Induction of accelerated blood clearance (ABC) phenomenon by PEG results from anti-PEG IgM produced after the initial injection [113]. Injections of siRNA complexed with PEGylated cationic liposomes (PLpx) can lead to rapid clearance of subsequent doses of PLpx by inducing production of anti-PEG IgM from peritoneal PEG-specific B cells [114]. Anti-PEG IgG and IgM are responsible for loss of efficacy of mRNA-LNP vaccines by inducing ABC phenomenon [115]. LNPs can activate splenic marginal zone B (MZB) cells to induce production of anti-PEG IgM, which in turn can activate the complement system and lead to the ABC phenomenon. Anti-PEG IgM can destabilize mRNA-LNP by releasing mRNA in a complement-dependent way [116]. Complement fragment C3 produced after activation of complement proteases can bind to complement receptors (CRs) expressed on phagocytes, which in turn can mediate the ABC phenomenon [110,117]. C3 convertase (C4b2a) can promote the cleavage of C3 into C3a and C3b, which can promote phagocytosis [118]. C5 convertase activated by C3 can promote the cleavage of C5 into C5a and C5b, which can promote formation of a membrane attack complex (MAC) to cause phagocytosis [118]. Figure 3 shows that PEGylated mRNA-LNP vaccine can induce production of anti-PEG antibodies, which in turn can promote complement activation to cause the ABC phenomenon.

An activated complement system can induce release of histamine and opsonization of LNPs [110]. This implies that activation of complement may lead to allergic inflammation. Mast cells, eosinophils, basophils, macrophages, plasmacytoid dendritic cells, and neutrophils all express receptors for C3a and C5a [119,120]. Anti-PEG IgM and IgG can induce production of anaphylatoxins C3a and C5a via the classical pathway, which in turn can induce degranulation of mast cells [121,122], smooth muscle contraction, and release of histamine, serotonin, PAF, and cysteinyl leukotrienes (CysLTs) [123,124]. C5a contributes to inflammation through activation of the NLRP3 inflammasome and induction of IL-1β [125]. C5a-C5aR signaling can increase numbers of Th1 and Th2 cells while decreasing the number of Treg cells in a mouse model of asthma [126]. C5 can prevent the development of experimental allergic asthma [127], meaning that complement activation can mediate allergic responses induced by mRNA-LNP vaccines. Figure 3 shows that anaphylatoxins can induce degranulation of mast cells to cause anaphylaxis. A better understanding of interactions between PEGylated nanoparticles and the blood immune system is necessary for developing effective and safe PEG-LNP vaccines to reduce ABC phenomenon and allergic responses.

## 8. How to Reduce Immune Responses Associated with PEG

Many polymers, including polyvinylpyrrolidone and poly(2-methyl-2-oxazoline), can induce polymer-specific antibodies [128]. These antibodies contribute to the pathogenesis of HSRs and the ABC phenomenon [129,130]. mRNA-LNP complexes can cause allergies and autoimmune diseases [90,131,132]. Thus, it is necessary to minimize these immune-related adverse reactions resulting from repeated delivery of mRNA-LNPs. Each component of LNP formulations can elicit an immune response. Replacing each component might reduce immune-related adverse reactions induced by mRNA-LNP vaccines. Since PEG is considered an epitope, PEG density on the surface of LNP can influence the production of anti-PEG IgM. High molecular weight PEGs can cause HSRs more easily than low molecular weight PEGs [133] owing to high level of anti-PEG IgM.

However, 1,2-distearoyl-sn-glycero-3-phosphoethanolamine-poly(ethylene glycol) (DSPE)-PE_2,40k_ (branched PEG lipid derivative) does not trigger the ABC phenomenon owing to the fact that 40 kDa DSPE-PE_2,40k_ can induce lower anti-PEG IgM levels than linear PEG-modified nanocarriers (DSPE-PE_40k_) [134]. Cleavable-branched and branched PEG-lipid derivatives can reduce the ABC phenomenon by inducing lower levels of anti-PEG antibodies [135].

High molecular weight (MW) free PEG can enhance the stability of PEG liposomes in animals with high levels of pre-existing anti-PEG antibody (APA) by suppressing APA production [136,137]. Free PEG can cause durable suppression of the APA response [136]. Free PEG might also be able to inhibit anaphylaxis by saturating B cell receptors (BCRs) to prevent the infused PEG-liposome from activating BCRs and inducing the production of APA. A lack of APA response might exert a negative effect on the ABC phenomenon.

Polysarcosine (PSar)-liposomes display fewer systemic and off-target interactions and higher stability than PEGylated liposomes [113]. PSar-liposomes can induce low levels of antibodies for PSar-liposomes [113]. Multiple injections of PSar-liposomes can reduce the ABC phenomenon by inhibiting accumulation of liposomes in the liver and spleen [113,138]. Liposomes with the longest PSar chain (68 mers) show enhanced stability and reduced ABC phenomenon compared with PEG-liposomes [139]. Poly (2-methyacryloyloxyethyl phosphorylcholine) (PMPC)-LNPs formulated from 1,2-dipalmitoyl-*sn*-glycero-3-phosphoethanolamine (DPPE)-LNPs display no cytotoxic effects or induce inflammatory responses. They also exhibit higher mRNA transfection efficiency than PEG-LNPs [108]. Poly-glutamic acid-ethylene oxide (PGE) graft copolymers can replace PEG on LNPs [83]. Unlike PEG-LNPs, PGE-LNP does not induce production of the anti-PEG antibody [83]. Injection of Rg3-liposomes does not induce production of IgM or activation of the complement system in blood circulation [140]. Thus, Rg3 can enhance the stability of liposomes without inducing the ABC phenomenon or immune responses. These reports indicate that replacing PEG with other polymers can decrease immune-related adverse reactions induced by PEG-mRNA-LNP vaccines.

Structural modification of the PEG moiety might reduce immunogenicity associated with PEG. Methoxy PEG grafting to allogeneic splenocytes can increase the number of Treg cells while decreasing Th17 lymphocytes [141]. This indicates that modification of PEG can induce immune tolerance. For example, polysialic acid-modified liposomes do not induce immune responses typical of PEGylated liposomes [142]. Inserting gangliosides into the bilayer beside PEG can also suppress the production of anti-PEG IgM [143]. A negative charge on PEG at the liposomal surface can induce HSRs by activating the complement system [144]. However, conjugating PEG200 to cholesterol (CHOL-PEG) can reduce complement activation to attenuate HSRs [144]. Thus, structural modifications of PEG can regulate immune-related adverse reactions by promoting immune tolerance.

## 9. Tolerance-Inducing mRNAs Can Reduce Allergies and Autoimmunity

Vaccine technology using mRNA-LNP can be employed to suppress antigen-specific immune responses [145,146]. Since mRNA-LNP vaccines can induce allergies and autoimmune response, it is critical to develop tolerogenic vaccines that are safe and effective. PEGylated nanoparticles can create a tolerogenic immune microenvironment by inducing lower complement activation than non-PEGylated PLGA nanoparticles around the injection site [147]. mRNA-LNP encoding non-allergenic MHC-II-binding epitopes has shown tolerogenic effects in a mouse model of peanut anaphylaxis [148]. This tolerogenic effect is mediated by tolerogenic Treg cells and accompanied by suppression of the production of Th2 cytokines (IL-4, IL-5, and IL-13), IgE synthesis, and increased expression of TGF-β and IL-10 [148]. PEG-mRNA-LNP enclosing antigenic T cell epitope can increase frequencies of Foxp3^+^Tregs while decreasing antigen-specific T cells producing TNF [149].

The PEGylated mRNA-LNP vaccine can be considered as a self-antigen that can cause autoimmune diseases. Modification of mRNA can treat allergies and autoimmune diseases associated with mRNA-LNP vaccines [148,150]. Modified mRNA-LNP can enter cells via endocytosis. After endocytosis, mRNA undergoes endosomal escape. mRNA is then translated into antigens. After proteasomal breakdown into peptides, these peptides are then presented on the surface of APCs to induce the generation of Foxp3^+^ regulatory T cells (Treg), which can induce tolerance to allergens or self-antigens [148].

Modifying nucleosides can suppress inflammation induced by exogenous mRNAs [151]. For example, uridine can be replaced with pseudouridine (Ψ) or m1Ψ, and cytosine can be replaced with 5-methyl cytosine. Unmodified mRNA can induce the production of type I IFN by stimulating innate immune sensors [152]. Unmodified ovalbumin (OVA)-LNP shows better anti-tumor effects than OVA-LNP with m1Ψ modification [153]. Unmodified OVA-LNP can increase the number of CD40^+^ DCs and the frequency of granzyme B^+^/IFN-γ^+^/TNF-α^+^ OVA peptide-specific CD8^+^ T cells [153]. Unmodified mRNA can also induce the production of proinflammatory cytokines including IFN-γ, TNF-α, and IL-2. However, high doses of Ψ-modified mRNA do not induce the production of proinflammatory cytokines [153,154]. Thus, Ψ-modified mRNA might induce a tolerogenic effect. It has been shown that m1Ψ-modified mRNA displays high translational efficiency without inducing innate immune responses [155,156]. Such m1Ψ-modified mRNA does not induce activation of TLR7 [157]. It has been found that m1Ψ mRNA can induce proliferation of immune suppressive FoxP3^+^ T reg cells to attenuate peanut-induced anaphylaxis. This effect is accompanied by decreased expression of Th2 cytokines and IgE synthesis [148]. However, m1Ψ mRNA does not lead to generalized immune suppression due to tolerogenic effects on APCs such as DCs [148]. In other words, DCs remain functional after internalization of m1Ψ mRNA.

Unmodified mRNA and small dsRNA can induce innate immunity and immune-related adverse reactions [158]. Modification and codon optimization of mRNA may decrease the risk of unwanted immune responses associated with mRNA-LNP vaccines. There are several ways to achieve immune tolerance: (1) nucleoside modifications, which may suppress activation of innate immune sensors and production of inflammatory cytokines; (2) optimization of UTR, which may increase translation efficiency and reduce immunogenicity of mRNA-LNP; (3) removal of dsRNA during IVT, which may reduce the production of inflammatory cytokines induced by mRNA-LNP vaccine; (4) reducing short RNA production during IVT, which may reduce the induction of innate immunity; and (5) introduction of conformational changes into T7 RNA polymerase, which may reduce production of dsRNA during IVT. Figure 4 shows that mRNA modification can induce immune tolerance to treat allergies, RA, and type I diabetes resulting from injection of PEGylated mRNA-LNP vaccines.

## 10. Discussion and Perspectives

Current clinical trials have shown that mRNA-LNP vaccines will continue to be treatment paradigms. However, these mRNA-LNP vaccines are known to cause unwanted immune responses. Vaccination with any COVID-19 vaccine (BNT162b2, Ad26.COV2.S, and mRNA-1273) can reduce risk of developing long COVID [159]. Individuals who have booster vaccination of mRNA-based COVID-19 vaccines (BNT162b2, mRNA-1273) display a higher risk of developing autoimmune connective diseases including rheumatoid arthritis and alopecia areata [160]. Concerns regarding the development of autoimmunity may exert a negative impact on uptake of COVID-19 vaccines. COVID-19 vaccines (BNT162b2 and CoronaVac) can cause autoimmune conditions requiring hospital care [161]. It would be necessary to develop tolerogenic mRNA-LNP vaccines to reduce the risk of developing autoimmune diseases.

The occurrence of PEG antibodies in the general population is increasing every year due to wide use of PEG in daily life and the development of sensitive assay systems for detecting anti-PEG antibodies. The ABC phenomenon induced by anti-PEG antibody could diminish the therapeutic efficacy of PEGylated products and increase the risk of HSR. It will be necessary to better understand the interplay between nanostructures and immune cells to reduce the ABC phenomenon. The contents of LNPs, dosing schedules, and routes of administration may reduce the ABC phenomenon. The role of PEGs in the induction of HSRs, ABC phenomenon, anaphylaxis, and mechanisms associated with these immunological reactions should be further investigated to make safe and effective mRNA-LNP vaccines. It is also necessary to test immunogenicity of PEGylated mRNA-LNP vaccines in animal models in association with the ABC phenomenon, CARPA, and HSRs.

To develop effective mRNA-LNP vaccines, it will be necessary to reduce unwanted immune responses associated with mRNA-LNPs. The following approaches may improve the efficacy of mRNA-LNPs: (1) optimizing mRNA sequence and molecular modification for inducing tolerogenic effects; (2) identifying route of administration for reducing immune-related adverse events; and (3) identifying PEG alternatives to reduce ABC phenomenon and HSRs. The induction of antigen-specific immune tolerance by modifying the mRNA-LNP vaccine has shown clinical benefits for treating allergic diseases. Circular IVT-mRNA produced by permuted intron-exon splicing circularization has shown low immunogenicity without altering protein production [162].

Since PEGylated LNP-mRNA vaccines can cause the ABC phenomenon and allergies, it will be necessary to identify PEG alternatives. Polypropylene glycol (PPG), polytetramethylene ether glycol (PTMEG), and poly-1,4-butylene adipate (PBA) can replace PEG in mRNA-LNPs [160]. However, these polymers may react with the anti-PEG antibody [163] and cause unwanted immune responses, including the ABC phenomenon, CARPA, and various other immune-related adverse reactions. The clinical skin prick test employing a panel consisting of PEG and PEG alternatives can enhance the chance to identify patients who might develop allergic reactions to mRNA-LNP vaccines containing PEG or PEG alternatives. To reduce the risk of the ABC phenomenon and CARPA associated with mRNA-LNP vaccines, it is necessary to identify individuals with natural anti-PEG antibody [78,98].

To develop mRNA-LNP vaccines that would not cause the ABC phenomenon or immune-related adverse reactions resulting from immunogenicity of PEG, it is necessary to understand interactions of PEG and PEG alternatives with immune cells. Studies on the immunogenicity of PEG and PEG alternatives should be carried out in the context of the whole PEG-mRNA-LNP vaccine. Public databases summarizing immunological and physicochemical properties of PEG and PEG alternatives using various excipients and whole products could provide valuable information on the selection of PEG and PEG alternatives for mRNA-LNP vaccines.

Since anti-PEG antibodies can be detected in individuals who have never received PEGylated vaccines, it is necessary to develop vaccines based on PEG-free delivery systems. Since PEG cannot be easily modified, it is reasonable to devise vaccines based on PEG-free systems. There have been reports concerning the effectiveness of PEG-lipid-free vaccines. It has been shown that a PEG-free two-component mRNA vaccine (PFTCmvac) containing a receptor binding domain of SARS-CoV2 can induce strong adaptive and innate immunity without causing immune-related adverse reactions [164]. Furthermore, PFTCmvac does not induce activation of the complement system [164]. Thus, PFTCmvac can be an alternative delivery vehicle for COVID-19 vaccines. mRNA-LNPs based on monoacyl POx/POz-lipids have displayed transfection efficiencies, cytocompatibility, and biophysical properties comparable to PEG-lipid equivalents [165]. PEG-lipid alternatives based on reversible addition-fragmentation chain transfer (RAFT) have displayed higher gene expression and antigen-specific antibody production than conventional PEGylated LNPs [166].

## 11. Conclusions

COVID-19 has accelerated the acceptance of mRNA-LNPs. mRNA-LNPs have shown clinical benefits. Transcriptomic analysis of individual patients will give valuable information for designing safe and effective COVID-19 vaccines. It has been shown that mRNA-LNPs can elicit unwanted immune reactions including anaphylaxis, hypersensitivity, and autoimmunity. To make more effective and safe vaccines, it will be necessary to develop tolerogenic mRNA-LNPs to prevent the induction of unwanted immune reactions. mRNA-LNPs have a tendency to become trapped in endosomes, which can exert toxic effects. It is necessary to improve endosomal escape of mRNA-LNPs to reduce toxicity and improve efficiency. Overcoming problems associated with mRNA-LNPs will make mRNA-LNPs a cornerstone of personalized therapy.

LNPs can carry Cas-9 mRNA or guide mRNA, suggesting that LNPs can be employed as a delivery vehicle for gene editing. In other words, mRNA-LNPs can be employed to treat various genetic diseases. To broaden the use of mRNA-LNPs, the targeting of mRNA-LNPs to various organs is needed. Based on recent technological progresses, the future of mRNA-LNPs vaccines beyond the pandemic is optimistic.

## Figures and Tables

**Figure 1 vaccines-13-00014-f001:**
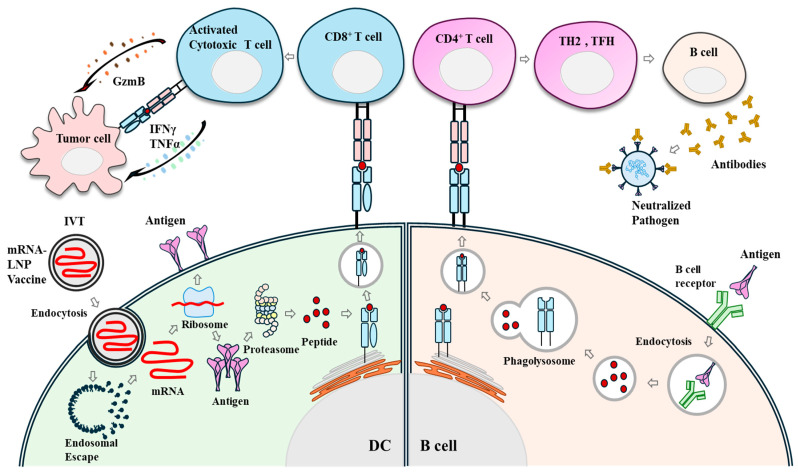
Immune responses induced by mRNA vaccines. mRNA-LNP vaccine is endocytosed by antigen presenting cells. Following endosomal escape, mRNAs are then translated into corresponding proteins. Proteins undergo proteasomal degradation. Peptides are then presented on MHC-I to induce activation of CD8^+^ T cells. Activated CD8^+^ T cells can secrete IFNγ and TNF-α. Activated CD8^+^ T cells can kill tumor cells by granzyme B (GzmB). Secreted proteins are recognized and engulfed by antigen presenting cells such as B cells. Antigenic peptides are presented on the MHC-II of B cells to induce activation of CD4^+^ T cells. Activated CD4^+^ T cells, such as TH2 cells and TFH cells, can activate B cells to induce production of antigen-specific antibodies. Arrows denote the direction of reaction. TFH denotes follicular T helper cells. IVT denotes in vitro transcription.

**Figure 2 vaccines-13-00014-f002:**
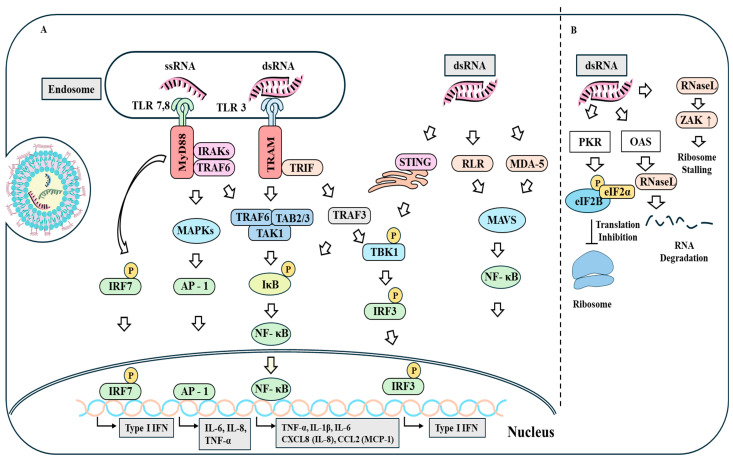
Innate immunity and translational inhibition induced by mRNA. (**A**) Endosomal TLR (TLR3, 7, or 8) can bind to ssRNA or dsRNA. Endosomal TLR7/8 can increase the production of type I IFNs and various proinflammatory cytokines including IL-6, CCL2, and CXCL10 by activating MYD88-NF-kB signaling. DsRNA produced during IVT can activate PKR or OAS pathway, resulting in degradation of mRNA. Endosomal TLR3 can bind to dsRNA and activate TRIF pathway. RLRs and MDA5 can bind to cytosolic RNAs and increase production of proinflammatory cytokines by activating IRF3 and N-kB. (**B**) DsRNA can bind to PKR and OAS. PKR can suppress mRNA translation by inducing phosphorylation of eIF2. OAS can activate RNase L, which in turn can induce translational arrest. Hollow arrows denote the direction of reaction. IRAK, interleukin-1-receptor-associated kinase; MAVS, mitochondrial antiviral-signaling protein; TAB, Mitogen-activated *protein* kinase kinase kinase 7-interacting *protein* 1; TAK, mitogen-activated *protein* kinase kinase kinase 7; TRAF6, tumor necrosis factor receptor-associated factor 6; TRAM, translocating- chain-associated membrane protein; ZAK, mixed lineage kinase. Hollow arrows denote the direction of reaction. T bars denote inhibition of reaction.

**Figure 3 vaccines-13-00014-f003:**
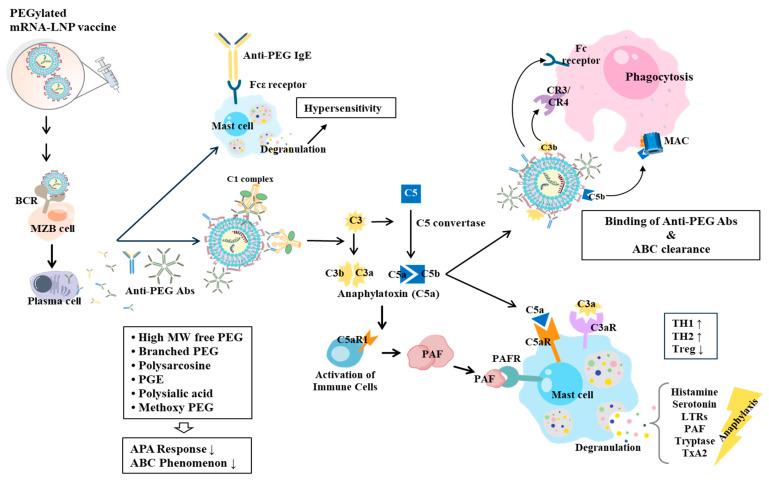
PEG induces complement activation-associated anaphylaxis and ABC phenomenon. PEGylated mRNA-LNP vaccine induces cross linking of B cell receptors. B cells are then differentiated into plasma cells. Plasma cells produce anti-PEG antibodies. Anti-PEG IgE activates immune cells such as mast cells and basophils, resulting in anaphylaxis through Fcε receptor in a complement-independent manner. PEG-anti-PEG-IgM complex can activate complement via classical pathway. Activation of complement produces anaphylatoxins such as C3a and C5a. These anaphylatoxins can stimulate mast cells to release various mediators and induce degranulation of mast cells. Anaphylatoxins can also induce immune cells to release PAF. PAF can bind to mast cells to cause degranulation of mast cells. PEG-mRNA-LNP vaccine can induce the production of anti-PEG IgM, which in turn can bind PEG-LNP. This binding can induce complement activation to produce complement fragments such as C3b and C5b. C3 can activate C5 convertase to produce C5a and C5b. These fragments can bind to complement receptors (CR3 and CR4) to induce ABC phenomenon by macrophages. Arrows denote the direction of reaction. ↓ denotes decreased expression/activity and ↑ denotes increased expression/activity. LTR, leukotriene receptor; TxA2, thromboxane A2; PGE, poly glutamic acid ethylene oxide.

**Figure 4 vaccines-13-00014-f004:**
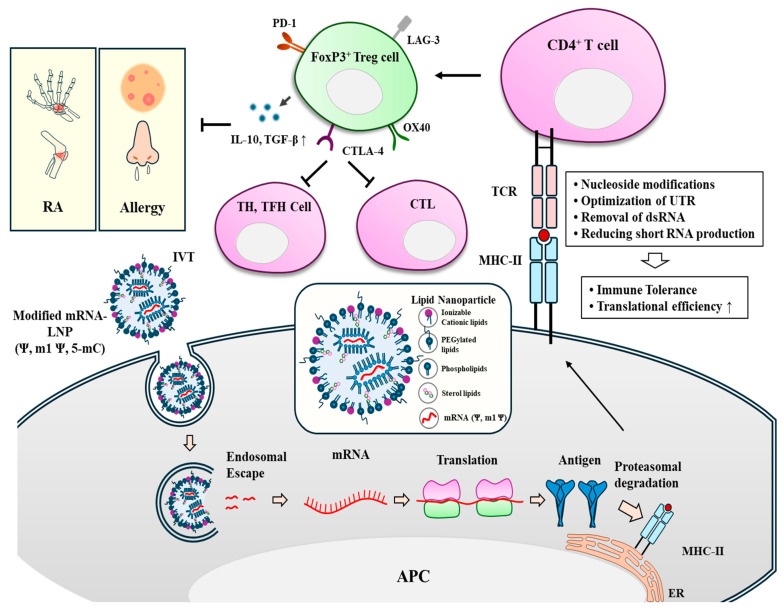
Induction of immune tolerance by mRNA-LNP. After endocytosis of modified PEG-mRNA-LNP vaccine by APC, mRNA undergoes endosomal escape. mRNA is then released and translated into antigen. Antigenic epitopes can be presented on MHC-II and bind to TCR on CD4^+^ T cells. CD4^+^ T cells are then differentiated into FoxP3^+^ Treg cells. Treg cells can secrete immune suppressive cytokines such as IL-10 and TGF-β. These Treg cells can induce immune tolerance. This tolerogenic effect is accompanied by decreased expression TH2 cytokines and IgE. CTLA-4, cytotoxic T-lymphocyte antigen 4; IVT, in vitro transcription; LAG3, lymphocyte activation gene-3; PD-1, programmed death-1; RA, rheumatoid. Arrows and hollow arrows denote the direction of reaction. T bars denote inhibition of reaction. ↑ denotes increased expression/activity.

## Data Availability

No new data were created in this study.

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
