# Peer review of "Regulating Immune Responses Induced by PEGylated Messenger RNA–Lipid Nanoparticle Vaccine"

_vaccines, 2024, doi:10.3390/vaccines13010014_

Round 1
Reviewer 1 Report
Comments and Suggestions for Authors
This review summarize the mechanisms of innate immunity induced by PEGylated mRNA-LNP vaccines and discuss the effects of modification or replacement of PEG on reducing immune responses of PEGylated nanoparticles. However, a similar review entitled “Immunogenicity of lipid nanoparticles and its impact on the efficacy of mRNA vaccines and therapeutics” has already been published elsewhere (PMID: 37779140). Thus, the addition of signature or latest literatures will add value to the manuscript. In addition, there are some comments that need to be addressed:
1. Please explain how free PEG improve the stability of PEG liposomes.
2. It is recommended to rearrange the logic in the second paragraph of Part 7. (1) Introduce the immunogenicity of PEGylated LNP is caused by anti-PEG antibodies; (2) Introduce the types of anti-PEG antibodies; (3) Provide examples of COVID-19 mRNA-LNP causing allergies through anti-PEG IgG/IgM. (4) These anti-PEG antibodies may reduce their effectiveness. (5) Remove the mechanism of immunity induced by anti-PEG antibodies to the next paragraph.
3. The sentence "Decreasing the PEG molarity or replacement of PEG structures can increase the protein expression [22]. This implies that PEG might affect innate and/or adaptive immun ity by regulating the production of antigenic epitopes during mRNA-LNP -promoted vac cination" cannot rule out the possibility that changes in protein expression levels are caused by changes in the physicochemical properties of formulations.
4. Some PEG-free mRNA vaccines aimed at reducing immune responses associated with PEG to make safe and effective vaccines (PMID: 36468425, 38918933, 39288840) should be discussed in the manuscript.
5. Abstract, Line 8, "pf particles" should be "of particles".
6. Page 2, Line 3, "intyerest" should be "interest".
7. Page 2, Line 3, "mRNA RNA" should be " mRNA".
8. Page 11, the tullstop is missing. "HSRs" should be "HSRs."
9. Figure 1, "MHCI" should be "MHC-I"; "MHCII" should be "MHC-II". Both are not labeled in Figure 1.
Comments on the Quality of English LanguageNA
Author Response
Dear Sir
Thank you for your excellent suggestions.
In this revision, I have made the changes to accommodate your suggestions. I add more references and change/rearrange sentences to make this manuscript more readable. I add conclusion section (lines 516-533). I also provide English certificate. I hope that the changes I have made are suitable.
Sincerely yours
Jeoung Dooil
Reviewer 1
Q1. This review summarize the mechanisms of innate immunity induced by PEGylated mRNA-LNP vaccines and discuss the effects of modification or replacement of PEG on reducing immune responses of PEGylated nanoparticles. However, a similar review entitled “Immunogenicity of lipid nanoparticles and its impact on the efficacy of mRNA vaccines and therapeutics” has already been published elsewhere (PMID: 37779140). Thus, the addition of signature or latest literatures will add value to the manuscript. In addition, there are some comments that need to be addressed:
Ans. Thanks. I agree. In this revision, I add more latest references to reflect current research trends. I add new references which show that COVID-19 vaccine can cause unwanted immune reactions including autoimmune disease (refs 159,160). I also add references which show that PEG-free mRNA-LNP vaccines can replace PEGylated mRNA-LNP vaccines (refs 164-166). I also rearrange sentences to make this manuscript more readable. Unlike PMID: 37779140, this manuscript mentioned the production of tolerogenic mRNA-LNP vaccines to prevent occurrence of allergy and autoimmune diseases. I also add conclusion section at the end of the manuscript. Conclusion section (516-533) suggests new applications (gene editing etc…) of mRNA-LNP vaccines beyond the pandemic.
Q2. Please explain how free PEG improve the stability of PEG liposomes.
Ans. Thanks. I agree. Please take look lines 351-356. It seems that Free PEG (high M.W.) enhances the stability of PEG liposomes by inhibiting the production of anti-PEG antibody (APA). APA is known to activate complement system, which in turn leads to the degradation of PEG liposomes. Therefore, free PEG can prevent adverse effects associated with PEG liposomes. PEG chains can also prevent macromolecules from degrading the liposome by creating shielding layers.
Q3. It is recommended to rearrange the logic in the second paragraph of Part 7. (1) Introduce the immunogenicity of PEGylated LNP is caused by anti-PEG antibodies; (2) Introduce the types of anti-PEG antibodies; (3) Provide examples of COVID-19 mRNA-LNP causing allergies through anti-PEG IgG/IgM. (4) These anti-PEG antibodies may reduce their effectiveness. (5) Remove the mechanism of immunity induced by anti-PEG antibodies to the next paragraph.
Ans. Thanks. I agree. I rearrange the second paragraph as you suggest. Please take look at lines 229-241. I also change the numbering or references.
Q4. The sentence "Decreasing the PEG molarity or replacement of PEG structures can increase the protein expression [22]. This implies that PEG might affect innate and/or adaptive immunity by regulating the production of antigenic epitopes during mRNA-LNP -promoted vaccination" cannot rule out the possibility that changes in protein expression levels are caused by changes in the physicochemical properties of formulations.
Ans. I agree. I change the sentences to make them more readable. Please take look at lines 97-103. Decreasing the PEG molarity or replacement of PEG structures can increase the size of nanoparticles, which in turn leads to more protein production.
Q5. Some PEG-free mRNA vaccines aimed at reducing immune responses associated with PEG to make safe and effective vaccines (PMID: 36468425, 38918933, 39288840) should be discussed in the manuscript.
Ans. Thanks. I agree with your suggestions. Please take look at lines 501-514. I add more references to make this manuscript more readable. In this revision, I add discussion based on the references (refs 164-166) you suggest (PMID: 36468425, 38918933, 39288840).
Q6. Abstract, Line 8, "pf particles" should be "of particles".
Ans. Thanks. I change it. Please take look at line 15.
Q7. Page 2, Line 3, "intyerest" should be "interest".
Ans. Thanks. I change it. Please take look at line 45.
Q8. Page 2, Line 3, "mRNA RNA" should be " mRNA".
Ans. Thanks. I change it. Please take look at line 45.
Q9. Page 11, the tullstop is missing. "HSRs" should be "HSRs."
Ans. Thanks. I change it. Please take look at line 470.
Q10. Figure 1, "MHCI" should be "MHC-I"; "MHCII" should be "MHC-II". Both are not labeled in Figure 1.
Ans. Thanks. I change it. Please take look at new figure legend 1.
Reviewer 2 Report
Comments and Suggestions for Authors
Dooil Jeoung et. al.,
Comments on the “Regulation of Immune responses induced by PEGylated mRNA-LNP vaccine” manuscript.
mRNA/LNP has rapidly grown as a central part of prophylactic vaccines and mRNA therapies. However, mRNA/LNP therapies need improvement, to say the least, for they are associated with unwanted adverse reactions including the induction of proinflammatory cytokines and chemotaxis, allergic reactions, potential cross reactivities with self-antigens, effects on BBB, chronic pathology, which need to be addressed. Examples of the clinical manifestation of such unwanted effects of mRNA-LNP are Long Haul COVID, effects on the hosts with autoimmunity, and allergies, to name a few. In this review manuscript, Dooil Jeoung and his colleagues present a well-thought and detailed and thorough summary and discussion of mRNA-based vaccines / mRNA-based therapy, with a focus on the advantages and the challenges of Polyethylene glycol (PEG) modifications. The review discusses the emerging ways to reduce unwanted immune responses associated with mRNA-LNP by i) notifications on mRNA, and ii) PEG structures or PEG alternatives.
The review covers, almost all, important steps / aspects of the design, biology, and immunopathology of LNP/mRNA and their pegylated platforms including mRNA transcription, structure of mRNA-LNPs, expression and presentation of corresponding antigens, the intrinsic adjuvant / immune-potentiating activities of mRNA and LNP moieties, mounting innate and adoptive immunity, innate RNA sensors, modifications of mRNA and their impacts on transcription, immunogenicity, and stability of mRNA.
1. Immune Response induced by mRNA Vaccines: Antigen Translation.
Lane 9 of the paragraph, there is an extra “.”, before “[4]”. Please, correct.
2. Structure of mRNA-LNPs.
Lane 13, “bloodstream” is written “blo1odstream”. Please, correct.
3. Reference 22 states that “Replacing DMG-PEG with DSG-PEG could slightly increase the protein expression”, the statement in the Line 10-11 of the second paragraph in the section 2 does not clearly explain the statement in the paper. Please, change the phrase to better explain the results of the reference “22”.
4. I suggest a discussion of intrinsic issues of mRNA based on actual clinical data, for example, i) post vaccination Long COVID, although vaccination reduces the risk of Long COVID, is important to be mentioned. ii) a risk of autoimmune diseases after COVID vaccination.
Below, I have cited a few references that help my point #4.
I am not suggesting including these references and I am not involved in these references.
Català, Martí et al. The effectiveness of COVID-19 vaccines to prevent long COVID symptoms: staggered cohort study of data from the UK, Spain, and Estonia
The Lancet Respiratory Medicine, Volume 12, Issue 3, 225 - 236
Jung, SW., Jeon, J.J., Kim, Y.H. et al. Long-term risk of autoimmune diseases after mRNA-based SARS-CoV2 vaccination in a Korean, nationwide, population-based cohort study. Nat Commun 15, 6181 (2024). https://doi.org/10.1038/s41467-024-50656-8.
Li X, Gao L, Tong X, Chan VKY, Chui CSL, Lai FTT, Wong CKH, Wan EYF, Chan EWY, Lau KK, Lau CS, Wong ICK. Autoimmune conditions following mRNA (BNT162b2) and inactivated (CoronaVac) COVID-19 vaccination: A descriptive cohort study among 1.1 million vaccinated people in Hong Kong. J Autoimmun. 2022 Jun;130:102830. doi: 10.1016/j.jaut.2022.102830. Epub 2022 Apr 14. PMID: 35461018; PMCID: PMC9008125.
Author Response
Reviewer 2
Dear Sir
Thank you for your excellent suggestions.
In this revision, I have made changes to accommodate your suggestions. I add more references and change the sentences to make this manuscript more readable. I add conclusion section. Conclusion section provides future direction of mRNA-LNP vaccines. I also provide English certificate. I hope that the changes I have made are suitable.
Sincerely yours
Jeoung Dooil
Comments on the “Regulation of Immune responses induced by PEGylated mRNA-LNP vaccine” manuscript.
mRNA/LNP has rapidly grown as a central part of prophylactic vaccines and mRNA therapies. However, mRNA/LNP therapies need improvement, to say the least, for they are associated with unwanted adverse reactions including the induction of proinflammatory cytokines and chemotaxis, allergic reactions, potential cross reactivities with self-antigens, effects on BBB, chronic pathology, which need to be addressed. Examples of the clinical manifestation of such unwanted effects of mRNA-LNP are Long Haul COVID, effects on the hosts with autoimmunity, and allergies, to name a few. In this review manuscript, Dooil Jeoung and his colleagues present a well-thought and detailed and thorough summary and discussion of mRNA-based vaccines / mRNA-based therapy, with a focus on the advantages and the challenges of Polyethylene glycol (PEG) modifications. The review discusses the emerging ways to reduce unwanted immune responses associated with mRNA-LNP by i) notifications on mRNA, and ii) PEG structures or PEG alternatives.
The review covers, almost all, important steps / aspects of the design, biology, and immunopathology of LNP/mRNA and their pegylated platforms including mRNA transcription, structure of mRNA-LNPs, expression and presentation of corresponding antigens, the intrinsic adjuvant / immune-potentiating activities of mRNA and LNP moieties, mounting innate and adoptive immunity, innate RNA sensors, modifications of mRNA and their impacts on transcription, immunogenicity, and stability of mRNA.
Q1. Immune Response induced by mRNA Vaccines: Antigen Translation.
Lane 9 of the paragraph, there is an extra “.”, before “[4]”. Please, correct.
Ans. I change it. Please take look at line 36.
Q2. Structure of mRNA-LNPs.
Lane 13, “bloodstream” is written “blo1odstream”. Please, correct.
Ans. I change it. Please take look at line 92.
Q3. Reference 22 states that “Replacing DMG-PEG with DSG-PEG could slightly increase the protein expression”, the statement in the Line 10-11 of the second paragraph in the section 2 does not clearly explain the statement in the paper. Please, change the phrase to better explain the results of the reference “22”.
Ans. Thanks. I change the phrase as you suggest. Please take look at lines 97-103.
Q4. I suggest a discussion of intrinsic issues of mRNA based on actual clinical data, for example, i) post vaccination Long COVID, although vaccination reduces the risk of Long COVID, is important to be mentioned. ii) a risk of autoimmune diseases after COVID vaccination.
Below, I have cited a few references that help my point #4.
I am not suggesting including these references and I am not involved in these references.
â‘ Català, Martí et al. The effectiveness of COVID-19 vaccines to prevent long COVID symptoms: staggered cohort study of data from the UK, Spain, and Estonia
The Lancet Respiratory Medicine, Volume 12, Issue 3, 225 - 236
â‘¡ Jung, SW., Jeon, J.J., Kim, Y.H. et al. Long-term risk of autoimmune diseases after mRNA-based SARS-CoV2 vaccination in a Korean, nationwide, population-based cohort study. Nat Commun 15, 6181 (2024). https://doi.org/10.1038/s41467-024-50656-8.
â‘¢ Li X, Gao L, Tong X, Chan VKY, Chui CSL, Lai FTT, Wong CKH, Wan EYF, Chan EWY, Lau KK, Lau CS, Wong ICK. Autoimmune conditions following mRNA (BNT162b2) and inactivated (CoronaVac) COVID-19 vaccination: A descriptive cohort study among 1.1 million vaccinated people in Hong Kong. J Autoimmun. 2022 Jun;130:102830. doi: 10.1016/j.jaut.2022.102830. Epub 2022 Apr 14. PMID: 35461018; PMCID: PMC9008125.
Ans. I appreciate your suggestion. I agree with your point. In this revision, I add more references to make this manuscript more readable. In this revision, I discuss potential of COVID-19 vaccines causing unwanted immune reactions such as autoimmune diseases. Please take look at lines 449-458. I add references 159 and 160 to accommodate your suggestion (risk of Long COVID and autoimmune diseases). Ref.161 also shows that COVID-19 vaccines can cause autoimmune diseases. I mention that COVID-19 vaccination can reduce the risk of developing long COVID (ref. 159, lines 450-451).